# MIMICKING HUMAN INTUITION: COGNITIVE BELIEF-DRIVEN Q-LEARNING

## ABSTRACT

Reinforcement learning encounters challenges in various environments related to robustness and explainability. Traditional Q-learning algorithms cannot effectively make decisions and utilize the historical learning experience. To overcome these limitations, we propose Cognitive Belief-Driven Q-Learning (CBDQ), which integrates subjective belief modeling into the Q-learning framework, enhancing decision-making accuracy by endowing agents with human-like learning and reasoning capabilities. Drawing inspiration from cognitive science, our method maintains a subjective belief distribution over the expectation of actions, leveraging a cluster-based subjective belief model that enables agents to reason about the potential probability associated with each decision. CBDQ effectively mitigates overestimated phenomena and optimizes decision-making policies by integrating historical experiences with current contextual information, mimicking the dynamics of human decision-making. We evaluate the proposed method on discrete control benchmark tasks in various complicate environments. The results demonstrate that CBDQ exhibits stronger adaptability, robustness, and human-like characteristics in handling these environments, outperforming other baselines. We hope this work will give researchers a fresh perspective on understanding and explaining Q-learning.

## 1 INTRODUCTION

Reinforcement learning (RL) algorithms aim to learn optimally rewarding behaviors by modeling how an agent acquires optimal strategies through a trial-and-error process within an environment (Sutton & Barto, 2018; Sutton et al., 1999). Although reinforcement learning has achieved significant success in areas like gaming, autonomous driving, and robotics, current algorithms continue to encounter challenges in addressing decision-making issues within complex, dynamic, and uncertain environments (Wu et al., 2024; McAleer et al., 2024; Xu et al., 2020; Watkins & Dayan, 1992; Silver et al., 2016; Mnih et al., 2015).

Q-learning, a cornerstone of model-free reinforcement learning (Watkins & Dayan, 1992; Watkins, 1989; Barto et al., 1989), along with its variants like Double Q Learning, has sought to improve learning by minimizing the mean squared Bellman error (MSBE). However, these methods often encounter challenges such as pessimistic value estimates and theoretical limitations (Ren et al., 2021; Hasselt, 2010; Hui et al., 2024), and they frequently fail to address the fundamental reliance on maximal value estimates (Fujimoto et al., 2018).

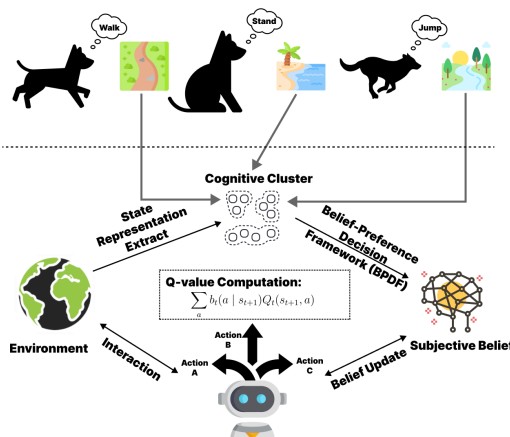

Figure 1: Cognitive Belief-Driven Q-Learning Framework: includes subjective belief components, human cognitive clusters, and BPDF. We provide a vivid example showing how pets make action decisions (e.g., walking, standing, jumping) in response to different environmental states (such as forest paths, oceans, and brooks).

To overcome these limits, we attempt to solve the problem using a novel approach: *Cognitive Science*, often seen as a manifestation of human intuition. In this domain, humans typically construct and adjust mental models' subjective beliefs when confronted with uncertainty to predict future events and make corresponding decisions (Peterson & Beach, 1967; Hastie & Dawes, 2009; Gigerenzer et al., 1991). These mental models, grounded in the cognition and experience of the world, empower humans to assess the potential consequences of various actions and make effective choices in complex settings. Notably, effectively managing uncertainty during decision-making is essential, as it directly influences both the efficiency of learning and the robustness of decisions (Kochenderfer, 2015). By leveraging this mechanism, we apply similar mental model theories to RL to improve the performance and adaptability of algorithms in various environments.

We present a novel direction for enhancing uncertainty optimization in deep Q-learning by integrating cognitive science's mental model with expected utility theory (Mongin, 1998). We propose Cognitive Belief-Driven Q-Learning (CBDQ), seen in Figure 1, an off-policy Deep Q-Learning algorithm applicable to both discrete and continuous states. Specifically, CBDQ incorporates:

(1) *Subjective Belief Component* (Soltani & Izquierdo, 2019) addresses the overestimation problem in Q-learning. It is grounded in Subjective Expected Utility Theory (Mongin, 1998), a fundamental component of decision theory that evaluates decision options by multiplying the utilities of actions by their associated probabilities. By modeling subjective beliefs, agents simulate how individuals adjust expectations, enhancing learning through probabilistic reasoning.

(2) *Human Cognitive Clusters*, implemented using the K-means algorithm (Ikotun et al., 2022), emulate how humans categorize information by grouping similar states within the environment's state space. This method mirrors human cognition, where stimuli or situations are naturally classified into distinct categories, and serves as an efficient tool for state representation extraction. The model compresses high-dimensional data by clustering the state space into meaningful, low-dimensional representations, capturing essential environmental features and reducing learning complexity.

(3) *Belief-Preference Decision Framework (BPDF)* integrates subjective beliefs and cognitive clusters into a unified decision-making process. BPDF adapts to various state spaces, allowing agents to base decisions on expected outcomes, past experiences (via Human Cognitive Clusters), and current beliefs. This enables context-sensitive decision-making, closely mirroring human cognition in complex, uncertain environments.

Empirical evaluations show that CBDQ consistently achieves higher feasible rewards in different environments, outperforming other advanced Q-learning baselines. This work moves us closer to human-like agents, offering innovative thinking for complex decision-making systems.

## 2 RELATED WORKS

The development of RL can be broadly categorized into two main directions, mathematical optimization and learning process simulation both stemming from the concept of learning from delayed rewards proposed by (Watkins, 1989).

### 2.1 ADVANCEMENTS MATHEMATICAL OPTIMIZATION IN Q-LEARNING

Despite efforts to address overestimation bias, Double Q-Learning (Hasselt, 2010) only partially reduces maximization bias and may still cause underestimation in noisy environments, potentially leading to convergence to near-optimal rather than optimal solutions (Weng et al., 2020; Ren et al., 2021). (Wang et al., 2021) proposed ensemble Q-learning as an alternative, using multiple Q-function approximators and conservatively selecting the minimum value. However, this strategy also risks underestimation and performance variability due to approximation errors and the limitations of a fixed ensemble size. In recent years, researchers have developed innovative Q-learning algorithms. For example, (Bas-Serrano et al., 2021) introduced Logistic Q-Learning, using a homoscedastic logistic noise model to reframe value learning via linear programming. (Garg et al., 2023) proposed Extreme Q-Learning (XQL), which utilizes a Gumbel noise source along with the LINEX loss function to more effectively capture the asymmetry in Q-value distributions. (Hui et al., 2023) developed Double Gumbel Q-Learning (DoubleGum), incorporating two heteroscedastic Gumbel noise sources and an adjustable pessimism factor to mitigate estimation bias. These approaches offer crucial theo-

retical and practical advancements for resolving Q-learning biases. While these optimization-based methods have partially addressed estimation bias, they remain incremental improvements within the Q-learning framework. Logistic Q-Learning has limited use in complex environments, XQL struggles with diverse uncertainties, and though DoubleGum offers a broader theoretical framework, it still faces key challenges, notably the lack of proven convergence. One might question: *Is there a unique way of thinking that can improve algorithms like Q-learning?*

## 2.2 LEARNING PROCESS INSIGHT ALGORITHMS IN REINFORCEMENT LEARNING

Ongoing development in human-like science and RL have increasingly focused on integrating human-like reasoning and beliefs, key components of learning process-oriented algorithms. These models aim to emulate human decision-making by adapting beliefs and strategies based on experience. Complementing these efforts, (Barber, 2012) discusses Bayesian reasoning frameworks that incorporate prior knowledge to manage uncertainty effectively. Building on this, (Carroll et al., 2019) explored collaboration by integrating learned human policies into Q-learning. More recently, (Zhang et al., 2021) introduced Solipsistic Reinforcement Learning, extracting human-perspective state representations, while (Hu et al., 2021) developed Off-Belief Learning (OBL), allowing agents to reason about others' actions with dynamic beliefs. Additionally, (O'Donoghue, 2021) proposed Variational Bayesian Reinforcement Learning, which offers a novel approach to balancing exploration and exploitation using a risk-seeking utility function. This method introduces a new Bellman operator with associated fixed points, termed 'knowledge values,' which compress both expected future rewards and epistemic uncertainty into a single value. These approaches enhance AI adaptability and align reinforcement learning with human cognition.

## 3 PROBLEM FORMULATION

**Markov Decision Processes (MDP)** To solve a RL problem, the agent optimizes the control policy under an MDP $\mathcal{M}$, which can be defined by a tuple $(\mathcal{S}, \mathcal{A}, p_{\mathcal{T}}, r, \mu_0, \gamma, T)$ where: 1) $\mathcal{S}$ and $\mathcal{A}$ denote the space of states and actions. 2) $p_{\mathcal{T}}(s_{t+1}|s_t, a_t)$ and $r(s_t, a_t)$ define the transition probability and reward function. 3) $\mu_0$ defines the initial state distribution. 4) $\gamma \in (0, 1)$ is the discount factor and $T$ defines the planning horizon. The goal of the RL policy $\pi(a|s)$ is to maximize expected discounted rewards:

$$\arg\max_{\pi} \mathbb{E}_{\pi, p_{\mathcal{T}}, \mu_0} \left[ \sum_{t=0}^{T} \gamma^t r(s_t, a_t) \right] \tag{1}$$

We define the action value function given a policy $\pi$:

$$Q(s, a) = \mathbb{E}_{\pi, p_{\mathcal{T}}, \mu_0} \left[ \sum_{t=0}^{T} \gamma^t r(s_t, a_t) \mid s_0 = s, a_0 = a \right] \tag{2}$$

and the optimal Q function is:

$$Q^*(s_t, a_t) = \mathbb{E}_{\pi, p_{\mathcal{T}}, \mu_0} \left[ r(s_t, a_t) + \gamma Q^*(s_{t+1}, a) \right] \tag{3}$$

One of our goals is that $Q$ is guaranteed to converge to $Q^*(s, a)$ as $t \to \infty$:

$$\lim_{t \to \infty} Q(s_t, a_t) = Q^*(s_t, a_t) \tag{4}$$

**Overestimation Error** Letting $Q(s_t, a_t; \phi_i)$ be the action-value function of Q-learning (Watkins & Dayan, 1992) at iteration i, we follow terminology from (Anschel et al., 2016). We denote $\hat{y}_{s,a}^i$ is the Q-learning target estimation, and $y_{s,a}^i$ is the true target:

$$\hat{y}_{s,a}^i = \mathbb{E}_{\mathcal{B}} \left[ r(s_t, a_t) + \gamma \max_a Q(s_{t+1}, a; \phi_{i-1}) | s_t, a_t \right], \tag{5}$$

$$y_{s,a}^i = \mathbb{E}_{\mathcal{B}} \left[ r(s_t, a_t) + \gamma \max_a (y_{s_{t+1},a}^{i-1}) | s_t, a_t \right]. \tag{6}$$

where $\mathcal{B}$ is a replay buffer. We denote $Z_{s_t,a_t}^i$ the target approximation error (TAE), and $R_{s_t,a_t}^{i,err}$ is the overestimation error, namely

$$Z_{s_t,a_t}^i = Q(s_t, a_t; \phi_i) - \hat{y}_{s_t,a_t}^i \tag{7}$$

$$R_{s_t,a_t}^{i,err} = \hat{y}_{s_t,a_t}^i - y_{s_t,a_t}^i \tag{8}$$

(Thrun & Schwartz, 2014) considered the TAE $Z_{s_t,a_t}^i$ as a random variable uniformly distributed in the interval $[-\epsilon, \epsilon]$. Due to the max operator in the target estimation $\hat{y}_{s_t,a_t}^i$, the expected overestimation errors $\mathbb{E}_z[R_{s_t,a_t}^{i,err}]$ are upper bounded by $\gamma\epsilon\frac{k-1}{k+1}$. $K$ is the number of actions. We attempt to overcome this overestimation issue with a unique approach and enhance the capabilities of Q-learning methods.

## 4 MODELLING SUBJECTIVE BELIEF DISTRIBUTION IN Q-LEARNING FRAMEWORK

In this work, we address a fundamental question: *How does integrating subjective beliefs refine decision-making within a Q-learning framework?* We propose a novel method, Cognitive Belief-Driven Q-Learning (CBDQ) to incorporate **human-like subjective belief components** into RL. By leveraging **Subjective Expected Utility Theory (SEUT)**, we dynamically update an agent's belief distribution over time, reflecting evolving perceptions of rewards, actions, and states.

### 4.1 EXPECTED UTILITY THEORY AND Q-LEARNING: A COGNITIVE PERSPECTIVE

To closely mirror human cognitive processes, we consider integrating SEUT into RL. SEUT offers a structured framework for decision-making under uncertainty by individual's belief preference, promoting actions that maximize the weighted sum of outcome utilities. This framework aligns seamlessly with MDPs, where the value function represents a specific form of expected utility derived from discounted returns.

**Proposition 4.1** *Consider a decision-making scenario in a MDP, where the complete set of possible outcomes is represented by $\mathcal{X}$. Let $b_t(\cdot \mid s_{t+1})$ represent the agent's belief distribution over possible actions in the next state $s_{t+1}$, and $u_t(s, x)$ be the utility of outcome $x$ in state $s$. Then the expected utility $U_t(s, x)$ at time $t$ is given by:*

$$U_t(s, x) = \sum_{x \in \mathcal{X}} b_t(\cdot \mid s_{t+1}) \cdot u_t(s, x) \tag{9}$$

Proposition 4.1 elucidates how individuals evaluate the utility of various actions within a MDP. It not only reflects the core tenets of SEUT but also provides a foundation for understanding learning processes. SEUT simulates how decision-makers assess potential outcomes through a weighted sum of utilities, which directly corresponds to the term $b_t(\cdot \mid s_{t+1}) \cdot u_t(s, x)$ in our formulation. The subjective belief component $b_t(\cdot \mid s_{t+1})$ represents an individual's belief, providing flexibility and robustness for modeling beliefs under uncertainty, aligning our model more closely with human cognitive processes. This characteristic aligns with the closely related cognitive processes proposed by (Tversky & Kahneman, 1992). Concurrently, research by (Hogarth & Einhorn, 1992) demonstrates that individuals revise their beliefs based on new information and experience.

### 4.2 EVOLVING BELIEFS IN Q-LEARNING

As outlined in proposition 4.1, the expected utility $U_t(s, a)$ in a MDP is computed from transition probabilities, rewards, etc. The CBDQ algorithm extends this by replacing the maximum Q-value update with a belief-weighted average of Q-values. We confirm that our Q function can converge to the $Q^*$.

**Theorem 4.1** *Given a finite MDP, the Cognitive Belief-Driven Q-Learning (CBDQ) algorithm, as given by the update rule:*

$$Q_{t+1}(s_t, a_t) = Q_t(s_t, a_t) + \alpha_t(s_t, a_t) \left[ r(s_t, a_t) + \gamma \sum_a b_t(a \mid s_{t+1}) Q_t(s_{t+1}, a) - Q_t(s_t, a_t) \right] \tag{10}$$

*converges with probability 1 to the optimal Q-function, as long as:*

$$\sum_t \alpha_t(s_t, a_t) = \infty, \quad \sum_t \alpha_t^2(s_t, a_t) < \infty \quad \text{for all } (s_t, a_t) \in \mathcal{S} \times \mathcal{A}. \tag{11}$$

To establish Theorem 4.1, we need an auxiliary result from stochastic approximation. You can check the convergence proof section in Appendix D.

It is important to note that while our method bears formal similarities to Expected SARSA, the introduced belief distribution $b_t(a \mid s_{t+1})$ fundamentally differs from the agent's actual action policy. $b_t(a \mid s_{t+1})$ represents the agent's subjective estimation of future states and rewards, influencing Q-value updates without directly determining action selection. The exploration policy (e.g., $\epsilon$-greedy) is responsible for action selection, ensuring comprehensive exploration of all state-action pairs. For algorithm convergence, $b_t(a \mid s_{t+1})$ must converge over time to selecting the action with the maximum Q-value, while the exploration policy maintains randomness to ensure non-zero probability of visiting all states. A parametric form for $b_t(a \mid s_{t+1})$ can be updated based on state transitions and rewards, similar to the probability smoothed Q-learning approach. (See Appendix A for more on the differences between Expected SARSA and CBDQ.)

Now we will demonstrate how CBDQ addresses the overestimation issue and introduce a lemma to assist us in solving this problem.

**Lemma 4.1** *Consider a MDP with state $s_{t+1}$ and actions $a$, along with Q-value estimates $\tilde{Q}_t(s_{t+1}, a)$, where $\tilde{Q}_t(s_{t+1}, a)$ is assumed to be unbiased for each $a$. Let $b_t(a \mid s_{t+1})$ denote the probability of selecting action $a$ in state $s_{t+1}$. By Jensen's inequality, for any convex function $f$ and random variable $X$, $\mathbb{E}[f(X)] \geq f(\mathbb{E}[X])$. Applying this to our setting yields:*

$$\sum_a b_t(a \mid s_{t+1}) \tilde{Q}_t(s_{t+1}, a) \leq \max_a \tilde{Q}_t(s_{t+1}, a) \tag{12}$$

Lemma 4.1 establishes the theoretical basis for using subjective belief probability distributions in Q-value updates. By incorporating a belief distribution, the target value $\sum_a b_t(a \mid s_{t+1}) Q_t(s_{t+1}, a)$ acts as a "downward estimate" of the maximum Q-value, reducing overestimation and improving the stability and reliability of Q-value updates.

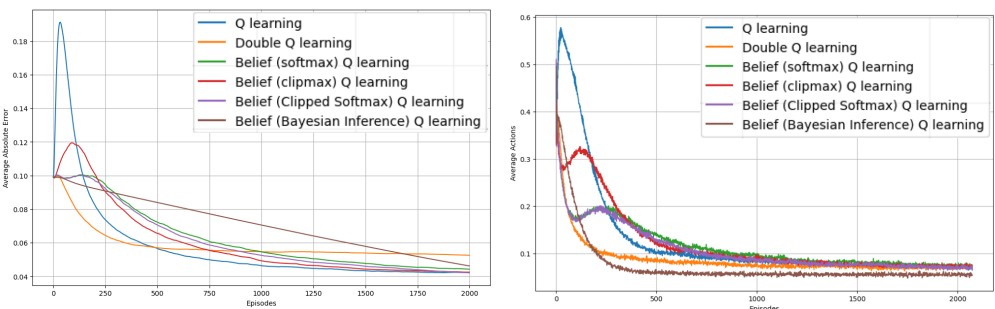

(a) Convergence of Belief Q-Learning vs Standard and Double Q-Learning

(b) Maximization Bias in Q-learning: Action Selection from Suboptimal States

Figure 2: Two key aspects of maximization bias in Q-learning and its variants. **(a)** compares the convergence of $|\tilde{Q} - Q^*|$ across belief Q-learning, standard Q-learning, and Double Q-learning. Belief Q-learning significantly reduces overestimation of Q-values while converging faster than Double Q-learning. **(b)** shows the fraction of times the suboptimal "Left" action is chosen from state A, demonstrating the effect of maximization bias in standard Q-learning.

We conducted experiments based on Example 6.7 in (Sutton & Barto, 2018)'s research (MBP) to verify the effectiveness of dynamically updating the subjective belief model. Four smoothing strategies, each employing a different fixed subjective belief probability model (Softmax, Clipped Max,

Clipped Softmax, and Bayesian Inference), detail in Appendix C are compared with Q-learning and Double Q-learning to demonstrate the universality and accuracy of the dynamic updating mechanism for managing uncertainty.

Figure 2 highlights differences in convergence speed and estimation bias across algorithms, with Belief Q-learning using Bayesian inference showing superior stability and convergence to the optimal value, underscoring the importance of dynamic belief updating and prior knowledge in decision-making (Barber, 2012).

Our studies suggest that relying solely on Q-values for probability models lacks robustness in diverse environments. Even Bayesian inference, while incorporating prior knowledge, is constrained by fixed distribution models. In contrast, human decision-making dynamically adjusts subjective belief probabilities based on accumulated experience, enabling better adaptation to complex and changing environments.

### 4.3 BELIEF INTERACTION AND UPDATE

Because of the limitations of fixed belief frameworks, we explore the application of dynamic beliefs from the perspective of learning processes. Figure 1 illustrates animals' subjective belief-based decision-making process in various contexts. This process reflects how agents simplify decision-making through state-space clustering, utilizing a strategy that groups states based on shared features (Liu et al., 2024).

To model belief interaction and update, we introduce **Belief-Preference Decision Framework (BPDF)**, which offers a structured approach to decision-making by integrating human prior knowledge with immediate belief updates. This framework enhances the efficiency and interpretability of decisions in complex environments. The model utilizes human expert knowledge to identify and select informative state features for representation learning. Additionally, clustering algorithms are applied to partition the state space $\mathcal{S}$ into $N$ semantically meaningful and internally consistent clusters $\{\mathcal{C}_n\}_{n=1}^N$, Figure 3 presents an example within the Box2D environment, adhering to the following formal criteria:

$$\mathcal{S} = \bigcup_{n=1}^N \mathcal{C}_n, \quad \mathcal{C}_i \cap \mathcal{C}_j = \emptyset, \forall i \neq j \quad (13)$$

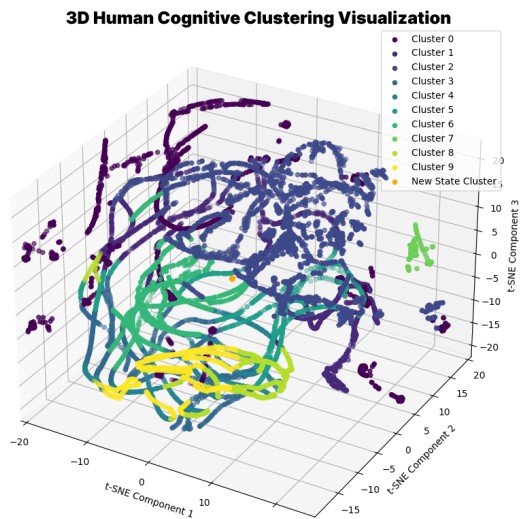

Figure 3: Cognitive Cluster Visualization for LunarLander. We utilized the t-SNE algorithm to map the high-dimensional state features into 3 dimensions. The orange points represent newly received states. If the closest cluster to them is Cluster 2, they will be automatically classified into Cluster 2.

Human cognition and belief formation are gradual processes, where early decisions rely on immediate rewards. Cognitive science research suggests that in uncertain environments, humans initially depend on short-term feedback, progressively incorporating long-term preferences as experience accumulates (Doya, 2007; Gershman et al., 2015). This shift from reward-driven choices to informed decisions underpins the dynamic belief framework we propose. The clusters in our model balance real-time beliefs with prior preferences, mirroring human cognition. This process ensures that, as the agent refines its beliefs, action selection converges to the optimal one, guaranteeing maximum utility. To balance immediate beliefs and prior preferences, the BPDF model updates subjective belief distribution $b_t(a \mid s_{t+1})$:

$$b_t(a \mid s_{t+1}) = (1 - \beta_t) \cdot \hat{b}_t(a|s_{t+1}) + \beta_t \cdot p_k(a|s_{t+1}) \quad (14)$$

where $\beta_t \in [0, 1]$ is a time-varying weight parameter that balances the influence between $\hat{b}_t(a \mid s_{t+1})$, representing the smoothed immediate reward strategy, and $p_k(a \mid s_{t+1})$, which reflects the subjective belief distribution for action selection in state $s_{t+1}$. After executing each action $a_t$, the BPDF model records the state-action pair in the corresponding cluster $\mathcal{C}_k$ and updates $p_k(a|s_{t+1})$ accordingly. This iterative process allows the model to continuously refine its decision-making strategy by integrating newly acquired knowledge while leveraging prior beliefs. The BPDF records action choices within each state cluster $\mathcal{C}_k$, computing the action selection probability distribution $p_k(a|s_{t+1})$:

$$p_k(a|s_{t+1}) = \frac{f(a \mid s \in \mathcal{C}_k)}{\sum_{\tilde{a} \in \mathcal{A}} f(\tilde{a} \mid s \in \mathcal{C}_k)} \tag{15}$$

The clustering approach in our model, inspired by natural categorization mechanisms observed in human and animal cognition, plays a crucial role in extracting meaningful representations from complex state spaces (Botvinick et al., 2020; Rudin, 2019). This process, known as **conceptualization** or **categorization** in cognitive science, enables efficient deciding intricate environments by classifying similar states based on experience (Rosch & Mervis, 1975; Markman & Ross, 2003). Unlike models with fixed probability spaces, the dynamic belief updating mechanism optimizes decision-making by continuously adapting to changes, effectively compressing high-dimensional state spaces into manageable representations.

---

**Algorithm 1** Cognitive Belief-Driven Q-Learning Algorithm

---

**Input:** Q function $Q(s, a; \phi)$, target Q function $Q(s, a; \phi^-)$, learning rate $\alpha$, discount factor $\gamma$, running steps $T$, episodes $E$, replay buffer $\mathcal{B}$ and exploration probability $\epsilon$
**Output:** $Q^{CBDQ}(s, a; \phi_T)$
1: Initialize $Q(s, a; \phi)$ with random weights $\phi_0$;
2: Initialize replay buffer $\mathcal{B}$ with a fixed length;
3: Initialize Belief-Preference Decision Framework (BPDF) $\{\mathcal{C}_n\}_{n=1}^N$;
4: Initialize a $\epsilon$-greedy exploration procedure: Explore($\cdot$)
5: **for** $i = 0 ; i < E ; i++$ **do**
6:     Get initial state $s_0$ from the environment
7:     **for** $t = 0 ; t < T ; t++$ **do**
8:         Choose action $a_t$ using $\epsilon$-greedy: $a_t \sim \mathcal{U}(0, 1)$
9:         Execute $a_t$ to get reward $r(s_t, a_t)$, next state $s_{t+1}$
10:        Store $(s_t, a_t, r(s_t, a_t), s_{t+1})$ into $\mathcal{B}$
11:        Find the cognitive cluster $\mathcal{C}_i$ of $s_t$, update the count of $a_t$ in $\mathcal{C}_i$
12:        Sample $N$ tuples from $\mathcal{B}$ to update $Q$ function:
13:           $y_{s_t, a_t}^i = \mathbb{E}_{\mathcal{B}} \left[ r(s_t, a_t) + \gamma \sum_a b_t(a \mid s_{t+1}) Q(s_{t+1}, a; \phi^-) | s_t, a_t \right]$
14:           The computation of $b_t(a \mid s_{t+1})$ in Equation 14 dynamically integrates immediate rewards and subjective beliefs, enabling continuous adaptation based on evolving information.
15:           $Loss = \mathbb{E}_{\mathcal{B}} \left[ (y_{s_t, a_t}^i - Q(s_t, a_t; \phi))^2 \right]$
16:        Update $\phi^-$;
17:     **end for**
18: **end for**

---

## 5   Experiment

**Running Setting.** For a comprehensive comparison, we employ *Feasible Cumulative Rewards* metric, which calculates the total rewards accumulated by the agent across all environments (higher is better). We run experiments with three different seeds (123, 321, and 666) and present the mean ± std results for each algorithm. To ensure a fair comparison, we maintain the same settings and parameters for all baselines. Our code is implemented based on the XuanCe benchmark (Liu et al., 2023). Appendix E.4 reports the detailed parameters.

**Comparison Methods.** We consider CBDQ (Algorithm 1) alongside the following baselines: (1) **DQN** (Mnih et al., 2013) approximates the action-value function using a deep neural network, with

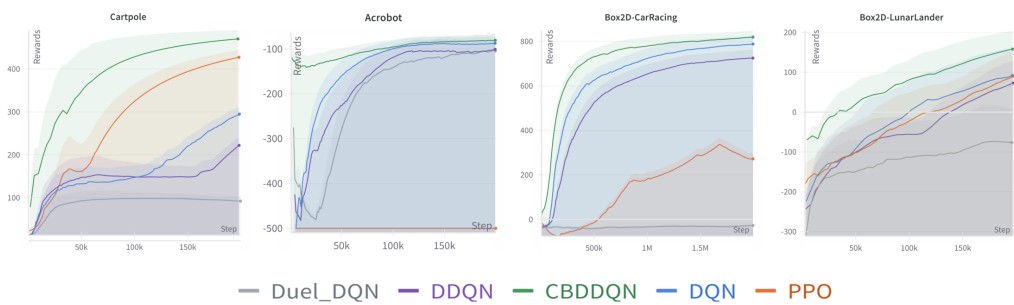

Figure 4: Feasible cumulative rewards. From left to right, the environments are Cartpole, CarRacing and LunarLander.

experience replay and target networks for stabilization. (2) **DDQN** improves on this by separating action selection from value estimation, reducing overestimation bias. (3) **DuelDQN** further enhances learning efficiency through a dual-stream architecture that individually estimates state values and action advantages. (4) **PPO** uses a clipped objective function for stable policy updates, balancing exploration and exploitation while maintaining a trust region for policy improvements.

## 5.1 EMPIRICAL EVALUATIONS IN PHYSICAL SIMULATION ENVIRONMENTS

The environments shown in Figure 4 and Appendix F highlight the performance of various RL algorithms across three distinct Classic Control and Box2D tasks (Towers et al., 2024; Parberry, 2017). The leftmost column displays the *Cartpole* environment, where agents are tasked with balancing a pole on a moving cart. Next is the *Acrobot* environment, where the goal is to swing a two-link arm to reach a specific height. The third column showcases the *CarRacing* task, a more complex scenario where agents must control a car to drive smoothly along a racetrack. Finally, the rightmost column presents the *LunarLander* environment, where agents must carefully land a spaceship on the moon's surface. Each environment progressively tests different control and decision-making skills, from balancing and swinging dynamics to managing more complex trajectories and landings.

Figure 4 illustrates CBDQ significantly significant improvements with faster convergence by leveraging subjective belief modeling and cognitive clustering. It outperforms all other approaches, generating stable, high-reward trajectories that closely resemble optimal policies. In contrast, without the BPDF, traditional Q algorithms struggle with slower convergence and lower final rewards. While PPO shows moderate improvements, it still suffers from inefficiencies in these environments.

## 5.2 EMPIRICAL EVALUATIONS IN COMPLEX TRAFFIC SCENARIOS

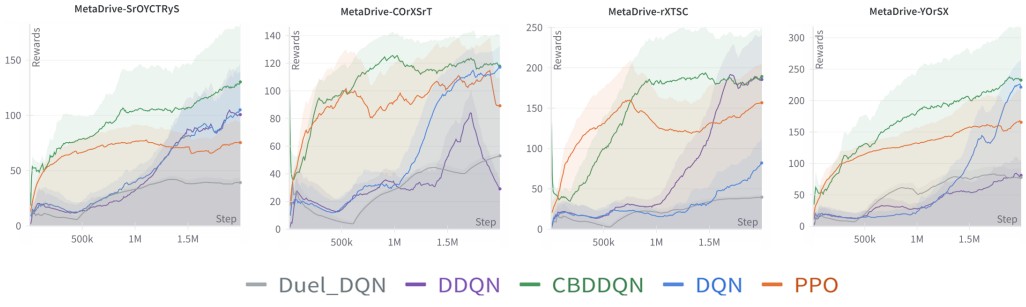

Figure 5: Feasible cumulative rewards. From left to right, the maps are SrOYCTRyS, COrXSrT, rXTSC, and YOrSX.

To evaluate the human-like decision-making and path-planning capabilities of our algorithm, we employ four complex environments within MetaDrive, each designed to mimic real-world driving

scenarios that require human-like adaptability (Li et al., 2022). Different letter combinations represent various types of road combinations. More detail of map design is in the Appendix.

Figure 5 and Appendix F present the obvious advantages of CBDQ, particularly in emulating human-like learning and decision-making. Compared to other algorithms, CBDQ demonstrates faster learning, greater stability, and superior final performance. Traditional Q-learning methods like Double DQN, Duel DQN, and DQN show significantly slower convergence and achieve lower rewards, indicating their limitations in handling the complexity of this environment. Unlike PPO, which often converges to suboptimal solutions, CBDQ's learning curve rises quickly and steadily improves, reflecting its ability to adapt and optimize in complex environments, avoiding local optima. Its strong adaptability to high-dimensional state spaces, dynamic obstacles, and varied road conditions mirrors human decision-making under uncertainty. The superior trajectory smoothness, intersection handling, and road structure adaptability of CBDQ underscore its progress in replicating human-like driving behavior.

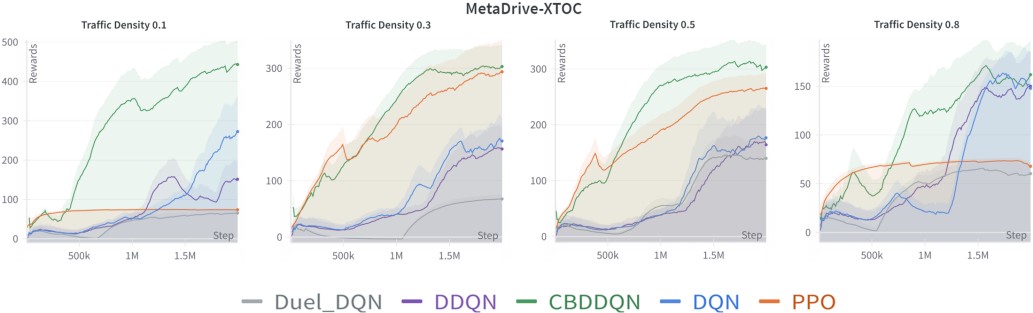

Figure 6: This figure compares the performance of different reinforcement learning algorithms under varying traffic densities (0.1, 0.3, 0.5, and 0.8) in the XTOC Map.

To assess driving control and decision-making at varying levels of difficulty, we conducted experiments with different traffic densities on the XTOC map. As traffic density increased, the system faced progressively complex challenges. Each sub-graph reflects the rewards obtained by agents as they learn to navigate through traffic at increasing levels of density.

Figure 6 and Appendix F highlight the superior performance of CBDQ across varying traffic densities, excelling particularly under high-density conditions. As traffic density increases, decision complexity grows, testing the system's ability to manage more intricate scenarios. While low-density traffic primarily challenges basic driving functions, high-density conditions require more complex decision-making and adaptive path adjustments. Leveraging the BPDF framework, CBDQ efficiently handles long-term planning, multi-lane interactions, and real-time risk management, consistently achieving higher reward values. PPO and traditional Q methods, though stable at moderate traffic densities, exhibit greater fluctuation in learning and decision-making under low- and high-density traffic, ultimately lagging behind CBDQ in both consistency and rewards.

In this experiment, we compare the performance of various algorithms under progressively increasing accident probabilities to evaluate their adaptability and decision-making capabilities in high-risk driving scenarios on the SSSC map (See Figure 7 and Appendix F). As the probability of accidents rises from 0.1 to 0.8, the complexity of the driving environment intensifies, requiring the algorithms to navigate regular driving challenges while also responding swiftly to sudden and unexpected risks. This setup tests the algorithms' ability to manage real-time dynamic environments, focusing on their long-term planning, risk avoidance, and decision stability under escalating uncertainty.

The experimental results indicate that CBDQ consistently outperforms other algorithms across all accident probability levels. At low and moderate accident rates, CBDQ demonstrates robust learning and stability, handling basic driving challenges while adapting efficiently to moderate risk scenarios. However, its advantage becomes more pronounced in high-risk environments, where accident probabilities reach 0.8. In these situations, CBDQ shows superior decision stability and maintains higher reward values compared to algorithms like PPO and DQN, which exhibit greater volatility and struggle to maintain performance as risks escalate. This highlights the strength of CBDQ's

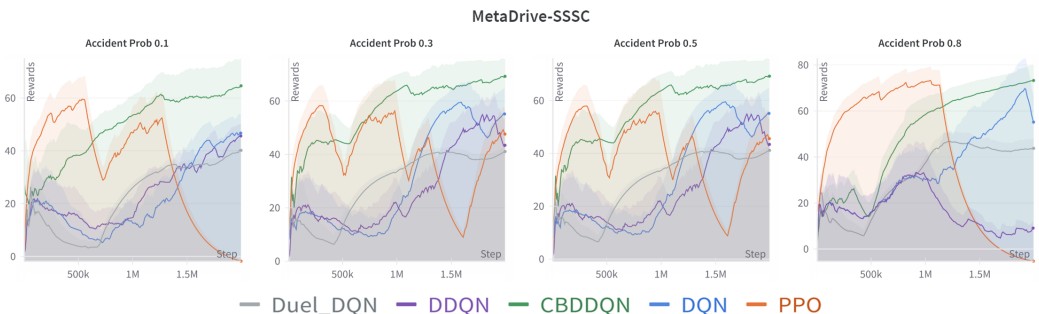

Figure 7: This figure compares the performance of different reinforcement learning algorithms under varying accident probability (0.1, 0.3, 0.5, and 0.8) in the SSSC Map.

belief-driven decision-making framework in navigating uncertainty and managing sudden hazards in dynamic driving environments.

## 6 FUTURE INSIGHT

**Expanding to Continuous Control Domains.** Building on our success in discrete environments, we are exploring ways to adapt our framework to continuous control scenarios. This involves integrating cognitive science principles with advanced reinforcement learning techniques, aiming for more flexible and robust decision-making in complex, continuous action spaces.

**Human-like Learning Processes in Reinforcement Learning.** CBDQ provides new insights for future reinforcement learning, particularly in emulating human learning processes. Future algorithms are expected to increasingly simulate human concept formation and abstract reasoning, with cognitive clustering evolving into autonomously formed conceptual hierarchies. Additionally, dynamic belief updating mechanisms point toward adaptive learning rates and exploration strategies, where algorithms adjust based on task complexity and learning progress. CBDQ's strengths in uncertainty management and long-term planning suggest that human decision psychology will play a greater role in future reinforcement learning.

## 7 CONCLUSION

This study introduces the Cognitive Belief-Driven Q-learning (CBDQ) algorithm, integrating cognitive science principles with reinforcement learning to enhance efficiency and interpretability in complex environments. CBDQ incorporates subjective belief probabilistic reasoning and cognitive clustering for state space representation, demonstrating superior performance over traditional Q-learning and advanced algorithms like PPO. This research has broad implications for AI, potentially catalyzing interdisciplinary innovations toward more intelligent, interpretable, and adaptable systems capable of interesting environments.

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
