# OpenReview forum: "Mimicking Human Intuition: Cognitive Belief-Driven Q-Learning"
_ICLR.cc/2025/Conference — ICLR 2025 Conference Withdrawn Submission_

### Official Review · Reviewer_giKZ · 2024-10-27

**Soundness:** 2
**Presentation:** 2
**Contribution:** 1
**Rating:** 3
**Confidence:** 4

**Summary:**

This paper proposes Cognitive Belief-Driven Q-Learning (CBDQ), which takes inspiration from human cognition and attempts to improve Q-learning by introducing (1) a subjective belief model that models the action distribution given the state and (2) a state-space clustering procedure. The authors theoretically show that this method can alleviate the overestimation problem in vanilla Q-learning and empirically verify its effectiveness in classic control tasks and simulated driving tasks.

**Strengths:**

- The authors theoretically show that the proposed method can address the overestimation problem in Q-learning.
- Empirical results are decent.

**Weaknesses:**

The biggest problem with the paper is that the proposed method CBDQ, although the authors have provided extensive discussions in the paper, is almost identical to the expected Sarsa.
- As the authors noted, the update formula is almost the same as Sarsa. The authors claim that the main difference is that the introduced belief distribution $b(a|s_t)$ differs from the agent's actual policy as in Sarsa, but to me, this point is invalid: since the only difference is caused by the exploration strategy in the agent's policy such as $\epsilon$-greedy, one can guarantee that the agent's policy converges to the optimal policy as $b(a|s_t)$ does by simply setting such as $\epsilon = 1/t$. More importantly, such a difference is not essential and, in my opinion, is not really related to the title and the motivation of the paper, i.e., mimicking human subjectivity.
- There are also some factual errors in the comparison between CBDQ and Sarsa. In Table 1 in the appendix, the authors claim that CBDQ is off-policy while Sarsa is on-policy. However, the update formula for CBDQ (Eq. (10)) depends on the agent's current policy due to the existence of the belief distribution term $b(a|s_t)$, which evolves over time. The authors also claim that the suitable scenario of Sarsa is "quick policy evaluation" which is not accurate as Sarsa was originally proposed as a policy _learning_ (rather than policy evaluation) algorithm.

There are also many technical details missing in the paper:
- The authors do not include the details of how the belief distribution $b(a|s_t)$ is implemented. I only found a sentence in Line 230 that vaguely says "A parametric form for $b_t(a | s_{t+1})$ can be updated based on state transitions and rewards", and no details in Algorithm 1.
- The proposed method also relies on state-space clustering. Yet there are no details on how such a clustering is performed. I also have some concerns about the scalability of clustering to more complex and high-dimensional inputs.

**Questions:**

- What is the advantage of explicitly using state-space clustering, given that simply using neural networks can often produce meaningful state representations already?

Minor:
- Line 302 says that Figure 3 is the clustering result in Box2D, while the caption of Figure 3 states that it is for LunarLander.

---

### Official Review · Reviewer_HwfH · 2024-10-31

**Soundness:** 3
**Presentation:** 3
**Contribution:** 2
**Rating:** 5
**Confidence:** 3

**Summary:**

Summary:

This paper presents a novel Cognitive Belief-Driven Q-learning (CBDQ) method, which integrates subjective belief modeling from cognitive science into the DRL framework.
CBDQ uses a cluster-based subjective belief model for state space representation to enhance the policy's learning and reasoning capabilities and mitigate overestimation phenomena.
The experimental results demonstrate that CBDQ outperforms previous Q-learning and advanced algorithms like DuelDQN and PPO in various benchmark tasks.

**Strengths:**

Strengths

1. This paper's writing is exceptionally smooth and easy to understand. The motivation, related work, and methodology are articulated and compared, and an appendix offers additional details for further clarification.

2. The authors introduce specific methods and concepts from cognitive science into the Q-learning framework of deep reinforcement learning (DRL), proposing the novel Belief-Preference Decision Framework (BPDF).

3. The authors conducted evaluations based on various settings. The experimental results demonstrate that this approach surpasses previous Q-learning algorithms.

**Weaknesses:**

Weaknesses

1. According to Equation 15, this method cannot be applied to tasks within a continuous action space, which significantly limits the algorithm's applicability, such as in robot control.

2. When faced with a high-dimensional state space, such as when the input is an image, how can cluster C in the state space be effectively initialized in advance (as needed in the third step of the algorithm)?

3. Why do the update rules in Equation 10 and Step 13 of the algorithm table differ?

4. This method appears not to address the problem of Q-value overestimation explicitly. Equation 12 only ensures that the target Q-value is less than or equal to the maximum Q-value rather than the maximum Q-value minus some offset.
Furthermore, in Equation 14, the update of the b function depends on p(a|s_t+1).  Suppose the current policy mistakenly overestimates the Q-value of a certain action. In that case, the probability of selecting that action increases, subsequently causing the estimation of b() for that action to rise, which further exacerbates the overestimation of the Q-value.

5. How are the four different smoothing strategies incorporated into the algorithm, and which steps of the algorithm table do they correspond to?

6. If the main contribution of this method is its ability to suppress Q-value overestimation, is it possible for this method to be applied to some experiments in the offline DRL domain, such as the D4RL benchmark? In offline DRL, the issue of Q-value overestimation is quite severe, and numerous previous works have attempted to mitigate Q-value overestimation through various approaches, such as CQL[1] and TD3+BC[2].

7. There is a lack of ablation studies, such as results on more environments regarding the four different smoothing strategies. Moreover, Figure 2 does not specify the experimental environment. How is the state space S divided into N clusters, how is N predetermined, and does its variation affect the results?

8. The core idea of this method is very similar to Bayesian inference. Is there any underlying connection between them? Could the authors further elaborate on the similarities and differences between the two?

[1] Kumar, Aviral, et al. "Conservative q-learning for offline reinforcement learning." Advances in Neural Information Processing Systems 33 (2020): 1179-1191.

[2] Fujimoto, Scott, and Shixiang Shane Gu. "A minimalist approach to offline reinforcement learning." Advances in neural information processing systems 34 (2021): 20132-20145.

**Questions:**

Please check the weakness section.

---

### Official Review · Reviewer_phxd · 2024-10-31

**Soundness:** 2
**Presentation:** 2
**Contribution:** 2
**Rating:** 3
**Confidence:** 4

**Summary:**

This paper introduces a reinforcement learning algorithm called cognitive belief-driven Q-learning, which modifies the standard Bellman operator from:

$\mathcal{T}Q(s,a) = r + \gamma \max_a Q(s',a)  $

 to:

 $\mathcal{T}Q(s,a) = r + \gamma \sum_{a'}b(a'|s')Q(s', a')$,

where $b(a|s)$ denotes the agent’s belief distribution over possible actions and is dynamically updated according to the following rule:
$b(a'|s')= (1-\beta_t) \cdot \hat{b}(a'|s') + \beta_t \cdot p(a'|s')$.

where the belief distribution comprises three main components:

1. $\beta_t\in[0,1]$, a time-varying weight parameter,
2. $\hat{b}(a'|s')$, representing a smoothed immediate reward strategy
3. $p(a'|s')$, the subjective belief distribution for action selection in state $s'$

The novelty of the approach seems to reside in this belief distribution. However, the paper provides only a brief explanation (just above Algorithm 1, on page 7) about the update mechanisms for these three components, leaving the exact update rules unclear.

**Strengths:**

A better Bellman operator is indeed a valuable contribution to the reinforcement learning community, and in this regard, this paper represents a good attempt.

**Weaknesses:**

**(Minor) Organization & notation**. (1 organization) The paper’s organization could be improved, as the proposed algorithm (specifically, the update rule for the belief distribution) is introduced in Section 4.3, while its convergence analysis appears in Section 4.2. Presenting the convergence proof before fully describing the algorithm can be confusing for readers, as it requires understanding the update mechanism beforehand. (2 notation) I suggest using distinct symbols for the Q-function iteration, the belief distribution  b , and other related terms, rather than using $t$ solely to denote both timesteps and iterations. For instance, in Equation (4), the notation appears somewhat unprofessional: $\lim_{t\rightarrow\infty} Q(s_t, a_t) = Q^*(s_t,a_t)$

After carefully reviewing the convergence proof in Appendix D, I identified some potential flaws in the reasoning within the mean criterion (from Equation 7 to Equation 13):

**(Major) Unjustified assumption of belief distribution in proof D.1**.  The proof assumes that *the belief distribution places $(1-\delta_t)$ mass on the maximal state of  Q* (page 3 in appendix), yet there is no accompanying explanation or validation for this assumption (please correct me if I am mistaken). This explanation is crucial, as the absence of this condition could cause the algorithm to degrade to standard SARSA or another variant, thereby undermining the intended advantages of the proposed approach.

**(Major) Flawed use of assumption in proof D.1**. The statement *Since the Q is bounded and $E[F_t]$ converges to zero with probability 1, ….*.  appears to involve circular reasoning: by using  the desired conclusion as an implicit assumption. The aim of Proof D.1 is to demonstrate that the Bellman error (involving the new belief distribution  b ) defined as  $F_t = r_t + \gamma\sum_a b(a|s_{t+1}) Q_t(s_{t+1}, a) -Q_*(s_t, a_t))$ converges to zero. (forgive my abuse of the symbol t). Using this result as an assumption in the proof is problematic.

**Questions:**

This work is a good and interesting attempt to develop a better Bellman operator. However, the current version does not fully meet the standards of a solid contribution. I rate this paper as “reject, not good enough,” for the following reasons:

1. There appear to be significant issues in Proof D.1 regarding convergence (please correct me if I am mistaken).
2. The contribution of this work needs clarification. As noted, the Bellman operator and its variants are prone to either overestimation (as in Q-learning) or underestimation (as seen in double Q-learning and similar methods). If the proposed algorithm addresses the estimation bias issue, it would be essential to include not only the convergence analysis (after all, Q-learning converges well but suffers from overestimation) but also an analysis of the bias. This would provide a clearer picture of how the new approach balances or mitigates these issues.

---

### Official Review · Reviewer_DsAM · 2024-11-02

**Soundness:** 1
**Presentation:** 1
**Contribution:** 1
**Rating:** 1
**Confidence:** 2

**Summary:**

In this work the authors propose a belief-based Q learning which can relieve the over-estimation by downward estimation.

**Strengths:**

This work proposes a new way to relieve the over-estimation, which is to reduce the Q learning significantly on the high side.

**Weaknesses:**

First, the motivation of the work is mainly base on the neuroscience. Therefore, a formally and rigorous definition of the "subjective belief component" should be proposed. However, I only find an ambiguous description "the agent's belief distribution", so that I guess it should be a probability density function. Also, I cannot understand the BPDF. What does the "immediate reward strategy" mean? If the "subjective belief distribution" is somewhat frequencies? Does the cluster change or fix after initialization? Finally, an important hyper-parameter N, the cluster number, needs for a detail number and a detail analysis in the experiment.

Second, the presentation of this work is really hard to understand.

1) In this work, there are actually two $t$ in the equations, one represents the horizon of the trajectories, and the other represents the updating timeline. This is really confusing.

2) the definition of $b$ and $u$ are given far from the introduction, the definition of $\mathcal{C}$ is not clear, and the definition of letter $f$ and $\tilde{b}$ introduced in Eq. 15 are never given formally. I guess the $f$ should be the counter?

3) In the Algo. 1 the updating of $\phi^-$ and $\phi$ is ambiguous.

Third, if the Lemma 4.1 is based on the convex function and the Jensen's inequality? If so, how could a complex neural network be convex?

**Questions:**

N/A

---

### Note · Authors · 2024-11-24

I have read and agree with the venue's withdrawal policy on behalf of myself and my co-authors.